# Agmatine Mitigates Inflammation-Related Oxidative Stress in BV-2 Cells by Inducing a Pre-Adaptive Response

**DOI:** 10.3390/ijms23073561

**Published:** 2022-03-24

**Authors:** Katarina Milosevic, Ivana Stevanovic, Iva D. Bozic, Ana Milosevic, Marija M. Janjic, Danijela Laketa, Ivana Bjelobaba, Irena Lavrnja, Danijela Savic

**Affiliations:** 1Institute for Biological Research “Siniša Stanković”—National Institute of Republic of Serbia, University of Belgrade, 11000 Belgrade, Serbia; katarina.tesovic@ibiss.bg.ac.rs (K.M.); iva.bozic@ibiss.bg.ac.rs (I.D.B.); ana.milosevic@ibiss.bg.ac.rs (A.M.); marija.janjic@ibiss.bg.ac.rs (M.M.J.); ivana.bjelobaba@ibiss.bg.ac.rs (I.B.); irenam@ibiss.bg.ac.rs (I.L.); 2Institute of Medical Research Belgrade, Medical Faculty of Military Medical Academy, University of Defense, 11000 Belgrade, Serbia; ivanav13@yahoo.ca; 3Department for General Physiology and Biophysics, Faculty of Biology, University of Belgrade, 11000 Belgrade, Serbia; danijela@bio.bg.ac.rs

**Keywords:** microglia, oxidative stress, inflammation, adaptive stress response, agmatine

## Abstract

Neuroinflammation and microglial activation, common components of most neurodegenerative diseases, can be imitated in vitro by challenging microglia cells with Lps. We here aimed to evaluate the effects of agmatine pretreatment on Lps-induced oxidative stress in a mouse microglial BV-2 cell line. Our findings show that agmatine suppresses nitrosative and oxidative burst in Lps-stimulated microglia by reducing iNOS and XO activity and decreasing O_2_^−^ levels, arresting lipid peroxidation, increasing total glutathione content, and preserving GR and CAT activity. In accordance with these results, agmatine suppresses inflammatory NF-kB, and stimulates antioxidant Nrf2 pathway, resulting in decreased TNF, IL-1 beta, and IL-6 release, and reduced iNOS and COX-2 levels. Together with increased ARG1, CD206 and HO-1 levels, our results imply that, in inflammatory conditions, agmatine pushes microglia towards an anti-inflammatory phenotype. Interestingly, we also discovered that agmatine alone increases lipid peroxidation end product levels, induces Nrf2 activation, increases total glutathione content, and GPx activity. Thus, we hypothesize that some of the effects of agmatine, observed in activated microglia, may be mediated by induced oxidative stress and adaptive response, prior to Lps stimulation.

## 1. Introduction

Microglia, resident immune cells of the central nervous system (CNS), trigger neuroinflammation as a defense mechanism against internal/external threats. A neuroinflammatory event implies the activation of microglia and their transition from surveillant to phagocytic phenotype, accompanied by complex morphological and functional changes that result in induction of proinflammatory gene expression and, subsequently, in production of cytokines, large amounts of NO, and an outburst of ROS. Principally, microglial activation is a neuroprotective process. However, most neurodegenerative diseases are accompanied by reinforced duration and/or intensity of neuroinflammation, which contributes to disease pathology [1]. Therefore, managing this perpetual activation of microglia represents a potential therapeutic target.

Oxidative burst is a component of neuroinflammation, whereby microglia-generated ROS are directed against various pathogens or serve as secondary messengers implicated in both initiation and augmentation of microglial activation [2,3]. The main product of oxidative burst during microglial activation is O_2_^−^ generated extracellularly by membrane-bound NOX, specifically NOX2 that is predominantly expressed in microglial cells [4]. Although O_2_^−^ is mainly formed towards the extracellular space, intracellular production also follows microglial activation, with NOX family enzymes as contributors [5,6,7]. However, other intracellular sources of O_2_^−^, including the mitochondrial respiratory chain and enzymes such as COX, lipoxygenase, and XO, should not be neglected as they are induced upon stress or immune activation [8,9,10,11]. In addition, extracellular superoxide ‘backfires’ through dismutation and production of H_2_O_2_, which quickly enters the cell through specific aquaporins [12]. It is important to note that microglial cells are the second richest in iron content in the brain; therefore, H_2_O_2_, which potentially escaped detoxification by GPx or CAT, may produce reactive radicals known to affect lipids, such as the hydroxyl radical via the Fenton reaction [13]. Since microglia use oxidative burst as an anti-pathogen mechanism, they must protect themselves from oxidative stress and damage.

Two central regulators of inflammatory and antioxidative responses in microglial cells are redox-sensitive transcription factors NF-κB and Nrf2, respectively [14,15]. NF-κB is pivotal for proinflammatory gene expression and transition from the homeostatic state to the proinflammatory phenotype. ROS may be implicated in the activation of the kinases upstream of NF-κB that rapidly degrade the inhibitory complex between IκB and NF-κB, resulting in the release and translocation of NF-κB to the nucleus [3,16]. NF-κB is a major activator of iNOS promoter upon inflammatory stimuli. Induction of iNOS results in extended production of large NO quantities from its endogenous substrate arginine [17]. Such NO generation is associated with immunological defense mechanisms, but it may have a pathogenic role. The interplay between arginine metabolic pathways via iNOS or ARG1 is crucial in acquiring pro- or anti-inflammatory phenotypes in macrophages/microglia [18]. In response to NOX induction and increased ROS during microglial activation, cytoprotective mechanisms are primarily initiated via Nrf2 activation and transcription of genes regulated by antioxidant-response elements, including HO-1, an antioxidant enzyme implicated in immunomodulation [14,19,20]. Nrf2 induction represents a counterbalance to Lps-triggered NF-κB activation, reflected in the modulation of microglial phenotype and polarization towards anti-inflammatory state [20,21].

Agmatine is a biogenic amine, identified in mammals nearly 30 years ago. It is synthesized by L-arginine decarboxylation in the nervous tissue. Agmatine acts as a neurotransmitter/neuromodulator in the brain, involved in behavioral, visceral, neuroendocrine regulation, and learning and memory processes [22]. In this regard, exogenously applied agmatine positively affected several animal models of neurodegenerative and psychiatric disorders, including Alzheimer’s disease, Parkinson’s disease, cerebral ischemia, traumatic brain injury, depression, and anxiety (reviewed in [23]). Anti-inflammatory and antioxidant mechanisms have been reported to underlie the neuroprotective effects of agmatine, which is particularly important, bearing in mind that neuroinflammation is implicated in the pathology of the diseases listed above [24]. Therefore, microglial cells emerge as potential therapeutic targets of agmatine.

The inhibition of iNOS stands out as one of the main mechanisms of the anti-inflammatory effects of agmatine [25,26,27,28]. It was shown that agmatine could suppress NO production in activated microglia and elicit the antioxidant activity against Lps-induced ROS accumulation via Nrf2/HO-1 pathway in microglial peripheral counterparts—macrophages [29,30,31,32,33]. However, agmatine effects on pro-oxidative markers and the antioxidant protection system comprising glutathione, GR, GPx, CAT, and SOD in Lps-activated BV-2 cells have not been investigated. We explored if exogenous agmatine could simultaneously modulate microglia activation and redox status via NF-κB and Nrf2/HO-1 pathways and to mitigate Lps-induced proinflammatory markers and cytokines.

## 2. Results

### 2.1. Effects of Agm on Viability and Lps-Induced NO Production in BV-2 Cells

Three different dosages of Agm (1, 10, and 100 µm) did not affect cell viability regardless of the activation state of microglia (Figure 1a). The same experimental protocol was applied to determine the effects of Agm on Lps-induced NO production. A dose-dependent response was observed, with 100 µm Agm significantly reducing Lps-induced NO release (Figure 1b). Pertinent to this, Agm was further applied at 100 µm.

### 2.2. Effect of Agm on Arginine Catabolizing Enzymes: iNOS and ARG1 and Microglial Polarization

To estimate the impact of Agm pretreatment on the iNOS and ARG1 gene expressions and protein levels, q-PCR, Western blot analysis, and immunofluorescence microscopy were applied (Figure 2a,b). Beside iNOS and ARG1, COX-2 as a proinflammatory and CD206 as anti-inflammatory markers were analyzed (Figure 2c,d). Lps-induction of the *Nos2* gene is observed 4 h after stimulation and multiplied four times by the 6th h (*p* < 0.05). Agm pretreatment of Lps-stimulated cells did not affect *Nos2* gene expression although there was a tendency towards decrease, 6 h after Lps administration. Nevertheless, Agm pretreatment reduced the Lps-triggered increase in iNOS protein levels by 40% (Figure 2a). Agm did not affect *Arg1* gene expression either in non-stimulated or Lps-stimulated cells. Lps downregulated the *Arg1* expression at 6 h, but the ARG1 protein level (estimated after 24 h) in stimulated cells was unaffected compared to Ctrl. Interestingly, in Ctrl and Agm groups the expression of *Arg1* is significantly upregulated at 6th h compared with 4th h (*p* < 0.05). Time coincidence of *Nos2* upregulation under inflammatory and *Arg1* under homeostatic condition, probably represents the self-protection strategy of microglial cells since the competition between iNOS and ARG1 is relevant for the dichotomy of the microglial activation state (pro- vs. anti-inflammatory) [18]. Agm pretreatment of stimulated cells increased the abundance of ARG1 protein by 50% compared with control cells (Figure 2b). Although agmatine pretreatment did not influence Lps-induced *Ptgs2* expression, it significantly reduced the abundance of COX-2 protein triggered by Lps stimulation. However, Agm alone significantly increased COX-2 levels (Figure 2c). Regarding CD206, neither Lps nor Agm initiated statistically significant changes in the expression of the *Mrc1* gene at analyzed time points. In contrast, Agm pretreatment partially restored CD206 levels decreased by Lps (Figure 2d).

### 2.3. Agm Decreases Lps-Induced Production of O_2_^−^ and Activity of XO

Agm pretreatment did not alter Lps-induced *Cybb* expression, gene coding for NOX2 (Figure 3a). However, Agm showed inhibitory effects on O_2_^−^ generation in microglial cells 24 h after Lps stimulation, maintaining a higher level of this free radical than in control conditions (Figure 3b). Given that we used cell lysates as samples, registered production comes from multiple superoxide generators. In line with this, pretreatment with 100 µm Agm significantly suppressed Lps-enhanced XO activity (Figure 3c).

### 2.4. Agm Increases Biomarkers of Lipid Peroxidation

Depending on the activation status of the microglia, Agm (100 µm) showed a dual effect on the analyzed lipid peroxidation markers (Figure 3d–f). Namely, Agm, per se, significantly increased the endogenous level of 4-HNE adducts and MDA in non-stimulated cells. Agm administration before Lps stimulation prevented the full capacity of Lps in increasing these reactive aldehydes (Figure 3e,f). In addition, the percentage distribution of cells according to their average per-pixel fluorescence intensities of 4-HNE showed that the same range contained 90% of control cells, 45% of non-stimulated cells treated with Agm, 20% of Lps-stimulated cells, and 60% of Lps-stimulated cells pretreated with Agm (Appendix A).

### 2.5. Agm Enhances the Antioxidant Capacity of Non-Stimulated and Lps-Stimulated BV-2 Cells

The antioxidative potential of Agm in BV-2 cells was estimated via intracellular content of total glutathione, which is the most critical antioxidative defense system in microglia, and by determining antioxidative enzymes gene expressions, protein, and activity levels (Figure 4). Agm treatment of non-stimulated microglia increased the total glutathione content and GPx activity. The pretreatment of Lps-stimulated cells with Agm increased the glutathione level, GR level, activity, and CAT level and activity (Figure 4a,b,d). The effects of Agm cannot be attributed to altered gene expression but to post-translational regulation. According to the observed changes, pretreatment ‘fortified’ the defense of cells against general oxidation of intracellular targets by H_2_O_2_ and other oxidizing species, which are protected by reduced glutathione. Pretreatment of stimulated cells with Agm partially abolished Lps-induced reduction in GR and CAT activity (Figure 4b,d). Lps-triggered GPx and SOD1/SOD2 activity increase was unaffected by Agm pretreatment (Figure 4c,e).

### 2.6. Agm Prevents Lps-Induced NF-κB Translocation to the Nucleus

To evaluate the potential of Agm to affect the critical step in NF-κB signaling—nuclear translocation of the NF-κB p65 subunit, we used immunofluorescence and Western blot (Figure 5). Agm pretreatment prevented Lps-induced NF-κB translocation to the nucleus, evidenced by lowered fluorescence intensity levels (Figure 5a). The percentage distribution of cells according to their average per-pixel fluorescence intensities of nuclear p65 revealed that 90% of control BV-2 cells were in the range 20–140 AU, in contrast to 30% of Lps-stimulated cells (Appendix A). Administration of Agm to microglial cells alone or as pretreatment to Lps caused 80% of the cell population to belong to the same range (Appendix A). Immunofluorescence findings were in concordance with p65 Western blotting in nuclear and cytosolic extracts (Figure 5b,c). Agm reduced the level of the NF-κB p65 subunit in the nucleus triggered by Lps (Figure 5b). In all experimental groups, cytosolic expressions of the NF-κB p65 subunit were in line with data obtained from nuclear extracts (Figure 5c). Furthermore, Agm partially prevented the Lps-triggered degradation of IκB-α (Figure 5d).

### 2.7. Agm Increases Nrf2 Nuclear Content and Enhances HO-1 Expression

Here we evaluated whether Nrf2, the primary regulator of cytoprotective mechanisms upon oxidative stress, and its response gene HO-1, were induced upon agmatine treatment in basal and inflammatory conditions (Figure 6). Western blot analyses revealed that agmatine pretreatment of Lps-activated BV-2 cells significantly increased the nuclear Nrf2 protein level shortly after Lps application (Figure 6a). Agmatine-induced Nrf2 activation in Lps-stimulated cells resulted in significantly increased gene expression and protein levels of HO-1 (Figure 6b,c). Lps alone did not activate Nrf2 as early as agmatine, while upregulation of the *Hmox1* gene and an increased HO-1 protein level compared with control cells were recorded (Figure 6a–c). Agmatine alone showed significant late Nrf2 induction, 24 h after administration, verified by Western blot in whole cell lysates without nuclear and cytosolic separation (Figure 6d left). However, the nuclear location of Nrf2 was confirmed by immunofluorescence imaging (Figure 6d, middle). At the same time point, agmatine also increased Nrf2 protein levels under inflammatory conditions; furthermore, quantification of Nrf2 nucleus fluorescence concurred with Western blot analysis (Figure 6d right and left, respectively). In whole cell lysates, we identified Nrf2 as double bands at 100 kDa, with upper bands significantly augmented in Agm and LpsAgm groups. Together our data obtained from the Western blot and immunofluorescence analyses imply that Nrf2 upper bands are associated with nuclear localization.

### 2.8. Agm Alleviates Proinflammatory Cytokines Release from Lps-Activated Microglia

To confirm whether Agm displays anti-inflammatory properties by modulation of Lps-induced production of TNF, IL-1 beta, and IL-6, their gene expressions and release were measured (Figure 7a–c). Lps-induction of the investigated proinflammatory cytokines genes was observed 4 h after stimulation, and significantly changed by the 6th h for *Tnf* and *Il1b* in opposite directions (*p* < 0.05). Agm pretreatment reduced the levels of TNF (by 34%) and IL-6 (by 71%), which can be attributed to altered *Tnf* and *Il6* expressions 4 h after stimulation (Figure 7a,c). Although Agm pretreatment did not influence Lps-induced *Il1b* expression, it significantly reduced the release of IL-1 beta protein (by 52%), which might be the consequence of TNF decrease, since TNF is required for the secretion of several cytokines, including IL-1 beta (Figure 7b) [34].

## 3. Discussion

The baseline of our investigation is that Lps-generated ROS is intertwined with microglial proinflammatory activation, and that re-establishing the balance of the redox status may result in controlling the inflammation [14,15].

Our results imply that the agmatine-induced changes in the antioxidative system of activated BV-2 cells probably result from synergism between effects that are both independent and dependent on Lps stimulation. Autonomous agmatine effects included: induction of oxidative stress verified by increased markers of lipid peroxidation 4-HNE and MDA; delayed Nrf2 activation; increased total glutathione content; increased GPx and reduced GR activity; and increased COX-2 levels.

Although we treated cells with a non-toxic dosage of agmatine, 100 µm is relatively high. This dosage has already been linked to the induction of oxidative stress in rat liver mitochondria via increased H_2_O_2_ generation [35]. In line with this, our results show that agmatine alone increases GPx activity, pointing to removal of H_2_O_2_ and peroxide radicals.

We show here, for the first time, that agmatine-induced oxidative stress results in lipid peroxidation in non-activated microglia. Agmatine’s relation to lipid peroxidation was explored previously by measuring the MDA level in various in vivo systems, including CNS; however, none of the studies reported agmatine as an inducer of lipid peroxidation in control conditions [28,36,37,38]. MDA and 4-HNE are highly reactive aldehydes, which, due to their strong electrophilic nature, interact with a wide range of proteins and target multiple signaling pathways, inducing cellular damage and death [39]. Elevated MDA and 4-HNE levels have been identified in several neurodegenerative diseases [40].

On the other hand, endogenous production under physiological conditions is involved in cellular signaling associated with maintenance and survival [39,41]. The outcome of 4-HNE action is concentration-dependent [41]. Sublethal levels of 4-HNE elicit an adaptive stress response via activation of the Nrf2 and increased HO-1 expression in various cells, including members of the mononuclear phagocyte system [42,43,44,45,46]. In line with this, exogenously applied non-toxic concentrations of 4-HNE protect BV-2 cells against Lps-induced oxidative and nitrosative stress [46]. Furthermore, exposure to 4-HNE is associated with increased intracellular glutathione content due to post-translational modification of the enzyme involved in GSH biosynthesis—GCL, or via Nrf2 induced upregulation of GCL [47,48,49]. Agmatine exerts neuroprotective effects in vitro and in vivo due to the induction of Nrf2 and the subsequent increase in GCL expression under basal conditions, as well as corticosterone-induced stress [50,51]. In addition, agmatine was identified as an activator of the Nrf2/HO-1 pathway against Lps-induced stress in macrophages [32]. Our data support these findings as agmatine induces delayed Nrf2 nuclear overexpression simultaneously with increased total glutathione content under basal conditions. GSH is the immediate elimination pathway for electrophilic compounds by conjugation. Therefore, increasing levels of total glutathione above the basal level demonstrated here probably represent a self-protection strategy by BV-2 cells against increased levels of 4-HNE and MDA [52]. Importantly, cell viability is preserved, suggesting that agmatine-induced 4-HNE and MDA are not associated with direct cytotoxic effects. Overall, we interpret that the outcome of agmatine-induced lipid peroxidation in non-activated microglia is electrophilic stress primarily via 4-HNE, which activates Nrf2, and finally results in adaptive stress response and lessened toxicity of lipid peroxidation metabolites.

Furthermore, agmatine alone induces changes in the activity of the GSH recycling enzyme—GR and the level of COX-2, probably as a consequence of 4-HNE action. Namely, decreased activity of GR may result from product inhibition, as shown in macrophages in the context of electrophile-induced upregulation of GSH [53]. Also, 4-HNE is a known inducer of COX-2 in macrophages [54,55]. It should be noted that the agmatine-induced COX-2 in BV-2 cells is significantly above basal (1.5 times) but far below Lps induction (4 times). Moreover, it is reasonable to suggest that this induction of COX-2 by agmatine does not trigger an inflammatory response since other analyzed markers of microglia proinflammatory phenotype were similar to control conditions.

Substantial Lps induction of *Cybb*, which is the primary source of superoxide during proinflammatory microglial activation, is unaffected by agmatine treatment. Nevertheless, increased XO activity and XO-generated ROS are also relevant for the Lps activation of macrophages, while the inhibition of XO activity has been shown as beneficial in animal models of central and peripheral inflammation [9,56,57]. Here we show, for the first time, that agmatine-induced inhibition of XO activity coincides with the decrease in O_2_^−^ levels and propose that XO has a significant role in the Lps-triggered production of intracellular superoxide in microglia.

Inducible NOS and COX-2 are two other enzymes known for cross-regulation and overexpression during neuroinflammation [17]. Previous work on macrophages/microglia reports both stimulation and inhibition of COX-2 activity and expression by iNOS and its downstream products: NO and ONOO− [17,58,59]. Here, we join the body of evidence showing agmatine inhibition of iNOS activity in innate immune cells [29,30,31,32,33]. Except for acting directly on iNOS as a competitive inhibitor, agmatine reduces iNOS protein expression by suppressing NF-κB signaling, as shown here and in other studies [25,26,30,32,60,61], paralleled by reduction in COX-2 protein levels. Agmatine inhibition of iNOS and the consequent decrease in NO levels may underlie the observed drop in COX-2 protein levels. COX-2 is also a verified superoxide generator upon Lps stimulus in primary cortical neurons and the ischemic brain [62,63]. Therefore, agmatine modulation of COX-2 protein levels in an inflammatory setting, in addition to XO inhibition, may contribute to the drop in O_2_^−^ and limit the intracellular formation of highly toxic NO metabolite ONOO− (O_2_^−^ + NO → ONOO−).

Lps activation resulted in total glutathione pool depletion due to direct or enzymatically catalyzed scavenging of ROS and reactive nitrogen species (RNS), as expected [64,65,66]. It was shown that microglia respond to oxidative stress by increasing GPx expression [67]. In line with this and our previous findings [66], activated BV-2 cells show increased activity in GPx simultaneously with reduced CAT activity. It was shown that NO reversibly inhibits CAT activity [68]; therefore, robust iNOS induction by Lps and consequent NO generation are likely the causes of CAT inhibition observed in this study. H_2_O_2_ is considered as the primary ROS that activates NF-κB and mediates inflammatory gene expression in microglia upon proinflammatory stimuli [3,15]. Although we have not measured the intracellular H_2_O_2_, we expect that oxidative burst alongside three times lesser CAT activity in Lps-stimulated cells than homeostatic cells might create an H_2_O_2_ permissive environment, promoting its role as a mediator/amplifier of microglial activation. In this study, agmatine prevents Lps-induced NF-κB translocation to the nucleus due to the inhibition of IκB degradation, which agrees with previous reports [60,69,70]. In addition, the ability of agmatine to decrease iNOS activity and NO production in Lps-activated cells is probably the cause of CAT deinhibition, thus limiting H_2_O_2_ role as a signaling molecule in NF-kB activation.

Our results also support previous findings that in response to Lps stimulation, microglia enhance SOD2 expression [71,72]. Ishihara et al. revealed that NF-κB and activator protein 1 increased SOD2 transcription in activated microglia and proposed that negative feedback between NF-κB and SOD2 represents a mechanism that controls duration and strength of activation [71]. However, in our study, agmatine inhibition of NF-κB did not interfere with Lps-induced SOD2.

Agmatine exerts antioxidative action in various tissues, including CNS [28,36,38,73,74]. In the light of the results obtained here, we interpret the agmatine-induced oxidative stress in non-activated microglia as initiating the adaptive response via the Nrf2 and GSH systems, enabling them to cope with subsequent stressors, i.e., Lps-induced ROS. In favor of this, Lps-induced lipid peroxidation is significantly halted by agmatine pretreatment. Furthermore, agmatine readily increases nuclear Nrf2 protein levels after Lps stimulation, which enhances gene and protein expression of HO-1 and retains delayed Nrf2 nuclear overexpression also under proinflammatory conditions. As a result, agmatine increases total glutathione content in non-activated and activated microglia. In line with this, increased GSH content in BV-2 cells is associated with protection against Lps-induced activation [75].

Agmatine modulates the redox status of Lps-activated microglia and the master regulators of inflammatory and antioxidant responses, NF-κB and Nrf2, respectively, which is mirrored in changes of microglial activation phenotypes. Thus, agmatine enables the transition of pro- to anti-inflammatory microglia phenotype, confirmed by the suppressed release of TNF, IL-1 beta, and IL-6 and the reduced expression of iNOS and COX-2, concomitant to the increased expression of ARG1 and CD206. The same concept of anti-inflammatory action was recently shown in BV-2 cells for phytochemicals from chestnuts verified by reduction of NF-κB, TNF, and IL-1 beta, induced by doses of Lps similar to this research [76]. Our study demonstrates the ability of agmatine to restrain both neuroinflammation and associated oxidative burst, which may be helpful in neurodegenerative diseases accompanied by overactivated microglia. We also identify agmatine as an inducer of adaptive stress response via lipid peroxidation metabolites. Considering that the CNS is highly enriched with lipids and involvement of MDA and 4-HNE in neurodegenerative diseases, further studies are needed to identify whether agmatine induction of oxidative stress and consequently of lipid peroxidation could be demonstrated in the in vivo setup, specifically in the brain. These analyses may add a new factor to the risk/benefit calculations in the proposition of agmatine as a CNS therapeutic agent.

## 4. Materials and Methods

### 4.1. Cell Culture and Treatment

All experiments were performed on the BV-2 murine microglial cell line obtained from Dr. Alba Minelli, Università degli Studi di Perugia, Perugia, Italy. BV-2 cells were cultured in a medium RPMI 1640 supplemented with 10% heat-inactivated fetal bovine serum, and 1% penicillin/streptomycin (all from Gibco, Thermo Fisher Scientific, Waltham, MA, USA) in a humidified incubator at 37 °C/5% CO_2_. When cell line reached 80–90 % confluence, the cells were detached with 0.1% trypsin-EDTA (Sigma-Aldrich, St. Louis, MO, USA) and seeded for experiments in the culture dishes. After overnight incubation, agmatine sulfate salt (Agm, Sigma-Aldrich, St. Louis, MO, USA) treatment for 30 min preceded stimulation with 1 μg/mL Lps (Escherichia coli serotype 026:B6; Sigma-Aldrich Labware, Munich, Germany). Agm was administered in three doses of 1, 10, and 100 µm for cell viability analysis and measurement of NO production, while in further experiments, only 100 µm Agm was tested. Upon Lps stimulation, experiments were terminated at different time points depending on the method/assay applied: (i) 30 min for NF-κB p65, IκB-α, and Nrf2 detection; (ii) 4 and 6 h for gene expression analysis; and (iii) 24 h for viability and NO production, pro-oxidative biomarkers (4-HNE, MDA, O_2_^−^, XO activity) levels, antioxidative non-enzymatic and enzymatic response levels and activity (total glutathione, GR, GPx, CAT and SOD), iNOS, ARG1, COX-2, CD206, Nrf2 and HO-1 protein levels, and TNF, IL-1 beta and IL-6 production. Experimental groups were as follows: untreated control cells (Ctrl); Agm treated cells (Agm); Lps-stimulated cells (Lps), and Lps-stimulated cells pretreated with Agm (LpsAgm). Different culture batches were used for repeated experiments.

### 4.2. Cell Viability Assay and Measurement of Nitric Oxide Release

A cell viability test based on reducing tetrazolium salt MTT (3-(4,5-dimethylthiazol-2-yl)-2,5-diphenyltetrazolium bromide) to purple formazan crystals was used. After a period of 24 h following the Lps stimulation, BV-2 cells were incubated in MTT solution (5 mg/mL) for 10 min at 37 °C. After the dilution of colored products in dimethyl sulfoxide, absorbance was measured at 492 nm using a microplate reader (Synergy H1M, BioTek Instruments Inc, Winooski, VT, USA). Cell viability was expressed as % of mean optical density (OD 492 nm) relative to control ± SEM, performed in five replicates per group from three independent experiments.

The colorimetric Griess method was used to indirectly determine NO released in cell culture supernatants, via measuring nitrite, which is a NO primary and stable product. Cell-free supernatant aliquots, collected 24 h after Lps stimulation, were mixed with equal volumes of Griess reagent (1:1 mixture of 1% sulfanilamide in distilled water and 0.1% N-(1-naphthyl)-ethylenediamine dihydrochloride in 2% H_3_PO_4_). After 15 min of incubation, nitrite ions formed a pink diazo dye with the Griess reagent. Spectrophotometric measurements were performed at 570 nm (Synergy H1M). The standard curve of known sodium nitrite concentrations served to calculate nitrite concentrations (µm) in the samples. Results represent mean ± SEM from three independent experiments performed in triplicate per group.

### 4.3. RNA Extraction, Reverse Transcription, and Real-Time Polymerase Chain Reaction

Total RNA from BV-2 cells was isolated using RNeasy Mini Kit (QIAGEN, Hilden, Germany), while concentrations were determined at 260 nm on a Nanophotometer^®^ N60 (IMPLEN, Munich, Germany). The ratios of optical densities (OD): OD260/OD280 and OD260/OD230 served as indicators of the qualities of the sample. cDNA synthesis was performed using a High Capacity cDNA Reverse Transcription Kit, and quantitative real time-PCR (qPCR) analysis with SYBR™ Green reagents on QuantStudioTM 3 Real-Time PCR System (all from Applied Biosystems by Thermo Fisher Scientific, Waltham, MA, USA). Primers of target genes are listed in Table 1, and the expression levels were evaluated by the comparative 2^−∆Ct^ method, using glyceraldehyde 3-phosphate dehydrogenase gene (*Gapdh*) as an internal control. Results are presented as mean ± SEM and were performed in 5 replicates per group from three independent experiments.

### 4.4. Western Blot Analysis

Cytosolic and nuclear extracts for detection of NF-κB p65, IκB-α, and Nrf2 were obtained by the Nuclear and Cytoplasmic Extraction Reagents kit (NE-PER, Thermo Fisher Scientific, Waltham, MA, USA). Protein extractions from whole cells were obtained by lysis in ice-cold Triton X-100 buffer saline enriched with protease and phosphatase inhibitors (Thermo Fisher Scientific). Cell lysates from five replicates were pooled in each experimental group. Protein concentrations were determined using the Micro BCATM protein assay kit (Thermo Scientific, Rockford, IL, USA). Samples (20 μg for total cellular proteins and 15 μg for cytosolic and nuclear extracts in each lane) were loaded onto 10% (except for iNOS, CD206, and Nrf2—7.5%) polyacrylamide gels and resolved by gel electrophoresis. After transfer to a PVDF membrane (Merck Millipore, Darmstadt, Germany), unspecific binding was blocked by incubating membranes in 5% bovine serum albumin (BSA, Sigma-Aldrich, Munich, Germany) in Tris-buffered saline with Tween-20 1 h at room temperature. Primary antibodies were applied overnight at 4 °C, followed by the incubation with secondary antibodies for 2 h at room temperature. Dilutions and specifications of primary and secondary antibodies are given in Table 2. Protein bands were visualized using chemiluminescence (Super Signal West Femto Maximum Sensitivity Substrate) on the iBright CL1500 Imaging System (both from Thermo Fisher Scientific). Densitometric analysis of protein bands was performed using an ImageQuant 5.2 software package and the relative quantity of proteins was determined by normalizing the optical density (OD) of the protein bands of interest to the OD of β-actin (proteins from the total cell lysates or cytosolic extraction) or lamin B (proteins from nuclear extraction) bands from the same lane. Data are presented as mean fold of control ± SEM, from three to four separate determinations.

### 4.5. Immunofluorescence Microscopy and Image Analysis

BV-2 cells were seeded on glass coverslips/chamber slides. After the treatment protocol, cells were fixed, washed, permeabilized with 0.25% Triton X-100 (Sigma-Aldrich, St. Louis, MO, USA), blocked with 3% BSA or with 10% normal donkey serum (NDS, Santa Cruz Biotechnology, Santa Cruz, CA, USA) for 30 min and incubated overnight (at 4 °C) with primary antibodies. The following day, fluorophore-labeled (Alexa-555 or Alexa-568) secondary antibodies were applied for 1 h at room temperature, protected from light. Dilutions and specifications of primary and secondary antibodies are given in Table 2. Nuclei were stained with Hoechst 33342 (Sigma-Aldrich, St. Louis, MO, USA), and coverslips/chamber slides were mounted with Mowiol (Merck Millipore, Darmstadt, Germany). Simultaneously, negative controls underwent the same procedure, with the omission of primary antibodies. Images were captured by a Zeiss Axiovert fluorescent microscope (Zeiss, Oberkochen, Germany), and to quantify the fluorescence intensity of 4-HNE, NF-κB p65, and Nrf2, an identical exposure time was applied to all experimental groups. The original images were changed to 8-bit grayscale type (pixel value range from 0 (black) to 255 (white) arbitrary units (AU)) and analyzed using AxioVision Rel. 4.9.1 software (4-HNE) and Image J software (NF-κB p65, Nrf2). 4-HNE stained cells were outlined by a freehand outline tool and mean per-pixel fluorescence intensity (MFI) was determined for each cell. The background was subtracted. MFI/cell was calculated for each image (139 × 104 µm) and ten images (containing approximately 200 cells) were analyzed per experimental group. Data are presented as MFI of 4-HNE/cell ± SEM and are representative of three separate determinations. For the quantification of NF-κB p65 and Nrf2 fluorescence intensity in the nucleus, an image with a blue channel (nuclei visualization) was converted to a binary image to outline nuclei automatically. Next, delineations of nuclei were transferred to the same image with a red channel that visualized target transcription factors. The mean per-pixel fluorescence intensity of NF-κB p65 and Nrf2 in the nucleus were measured in each cell, then the MFI/nucleus was calculated for each image (222 × 166 µm), and 14–17 images (containing approximately 200 cells) were analyzed per experimental group. Data are presented as MFI of NF-κB p65 and Nrf2/nucleus ± SEM and are representative of three separate determinations.

### 4.6. Spectrophotometric and Fluorometric Assays for Determination of Redox Status

Cells were collected in cold PBS by a scraper, sonicated on ice (3 × 5 s at 10 kHz, Bandelin Sonopuls HD 2070, Berlin, Germany), and centrifuged for 5 min at 15,000× *g*, at 4 °C. The protein content in supernatants was determined by the method used by Lowry [77]. Supernatants served for the analysis of the following parameters of redox status: O_2_^−^, MDA, and total glutathione content, the XO activity, and activity of antioxidative enzymes: GR, GPx, CAT and SOD, as previously described [66].

Summarized, the content of O_2_^−^ in the samples was determined by the reaction where a superoxide serves as a reductant of nitroblue tetrazolium (NBT, Sigma-Aldrich, Munich, Germany); the absorbance of the colored product was measured at 550 nm and data are given as µm of reduced NBT/mg protein.

As a lipid peroxidation marker, MDA was measured spectrophotometrically (at 532 nm) after incubating samples with thiobarbituric acid (TBA) reagent and forming a red product. The results are presented as nmol of MDA/mL.

The XO fluorometric assay (Cayman Chemical Company, Ann Arbor, MI, USA) is based on a multistep enzymatic reaction during which XO catalyzes the conversion of hypoxanthine to xanthine and then to uric acid, forming H_2_O_2_ as a byproduct that reacts with ADHP (10-acetyl-3,7-dihydroxyphenoxazine) and produces a highly fluorescent compound—resorufin (ex/em: 520/585 nm). XO activity of the samples was determined according to the manufacturer’s instructions, and the results are presented as µU/mg protein.

Total glutathione was determined by the DTNB-GSSG reductase recycling assay, which implies oxidation of GSH and formation of yellow derivative 5′-thio-2-nitrobenzoic acid (TNB), measurable at 412 nm, and concentrations of glutathione are presented as nmol/mL.

GR maintains intracellular glutathione by reducing glutathione disulfide in the presence of NADPH as electron donor, and GR activity was determined by monitoring NADPH oxidation, visualized as a decrease in NADPH autofluorescence intensity (ex/em: 360/460 nm). The unit of GR activity is defined as the number of µm of NADPH oxidized per min and is normalized to protein content (mU/mg protein).

GPx activity was measured indirectly by a coupled reaction with GR and subsequent consumption of NADPH, accompanied by a decrease in absorbance at 340 nm. The results are expressed as mU/mg protein.

CAT activity was determined by measuring the absorbance (at 405 nm) of a stable yellow complex formed in H_2_O_2_ and ammonium molybdate reactions. The unit of CAT activity represents µmol of H_2_O_2_ reduced per min and is normalized to a protein content of the sample (U/mg).

The spectrophotometric assay for measurement of total SOD activity was based on the ability of SOD to inhibit epinephrine auto-oxidation at alkaline pH. Epinephrine oxidation leads to pink-colored adrenochrome, and its rate is measurable at 480 nm. One unit of SOD activity is defined as the amount of enzyme required to inhibit 50% of epinephrine autooxidation, then normalized to protein content (U/mg). SOD2 activity was determined in the presence of 5 mM KCN that inhibits SOD1. SOD1 activity was determined as the difference between total SOD and SOD2 activity. Data collected from all assays are presented as mean ± SEM, performed in 3–5 replicates per group from three separate determinations.

### 4.7. Enzyme-Linked Immunosorbent Assay (ELISA) for Quantitative Detection of Proinflammatory Cytokines

The TNF, IL-1 beta, and IL-6 production levels were measured in cell-free supernatants with a kit comprising antibodies for capture and detection (eBioscience by Thermo Fisher Scientific, Waltham, MA, USA). Briefly, the ELISA plate was incubated overnight (at 4 °C) with a capture antibody (purified anti-mouse/rat TNF, Cat. no. 14-7423-81; anti-mouse IL-1 beta, Cat. no. 14-7012-68A, or anti-mouse IL-6 antibody, Cat. no. 14-7061-81). The next day, biotinylated anti-mouse/rat TNF (Cat. no. 13-7341-81), or anti-mouse IL-1 beta (Cat. no. 13-7112-68A) or anti-mouse IL-6 polyclonal antibody (Cat. no. 13-7062-81) were used for detection. After incubation with avidin-HRP (Cat. no. 18-4100-94, eBioscience by Thermo Fisher Scientific), tetramethylbenzidine (TMB) Substrate Solution (Invitrogen by Thermo Fisher Scientific) was added, and plates were read at 450 nm using the microplate reader (Synergy H1M). Recombinant mouse TNF (Cat. no. ab212073, Abcam, Cambridge, UK) or IL-1 beta (Cat. no. 39-8012-60) or IL-6 (Cat. no 39-8061-60, eBioscience by Thermo Fisher Scientific) served for generation of a standard curve and to calculate cytokine levels (pg/mL) in the samples using a 5-PL logistic curve. Data are presented as mean ± SEM, from three independent experiments performed in 3 replicates per group.

### 4.8. Statistical Analysis

Statistical analysis was performed using IBM SPSS 25 software (SPSS, http://www-01.ibm.com/software/uk/analytics/spss, accessed on 11 February 2022, RRID: SCR_002865) and GraphPad Prism 8 software (GraphPad Prism, http://www.graphpad.com, accessed on 11 February 2022, RRID: SCR_002798). If data met homogeneity of variance (Levene’s test) and Gaussian distribution (Kolmogorov–Smirnov normality test), the mean values between multiple groups were compared with two-way ANOVA followed by Bonferroni’s post hoc analysis or by *t*-tests for comparing two groups. Otherwise, nonparametric tests were used, a Kruskal–Wallis test followed by Dunn’s post hoc test for multiple groups comparisons, or a Mann–Whitney test for comparing the mean values between two groups. The significance of the difference in mean values was expressed for Agm, Lps, and LpsAgm groups compared to the Ctrl group, and LpsAgm compared with the Lps group, except in the case of viability and NO production where the mean of each group was compared with the mean of every other group. An α level of 0.05 for all statistical tests was used. Data are presented as mean ± SEM and were considered statistically significant for *p* < 0.05.

## Figures and Tables

**Figure 1 ijms-23-03561-f001:**
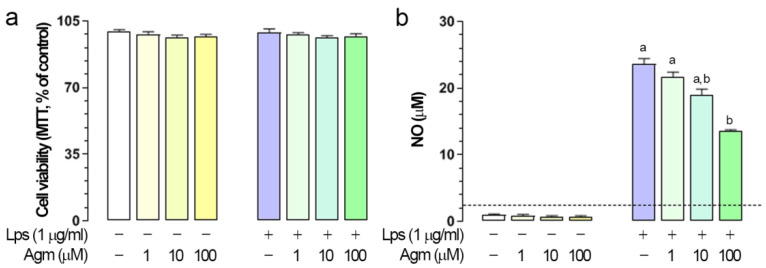
Agm effect on viability and NO production of control and Lps-stimulated BV-2 cells. If not stated otherwise, for these and analyses presented in the following figures, BV-2 cells were seeded at a density of 3.4 × 10^4^ cells/cm^2^. Cells were pretreated with agmatine sulfate—Agm (1, 10, and 100 µm) for 30 min and then stimulated with Lps (1 µg/mL) for additional 24 h. (**a**) Cell viability of control and Lps-stimulated BV-2 cells treated with Agm. (**b**) Modulation of Lps-induced production of NO with Agm. The dashed line represents the limit of detection of the Griess assay (~2.5 μm). Results are expressed as the mean ± SEM (n = 3) and were analyzed by Kruskal–Wallis test followed by Dunn’s multiple comparisons tests; the groups not sharing a joint letter are significantly different (*p* < 0.05).

**Figure 2 ijms-23-03561-f002:**
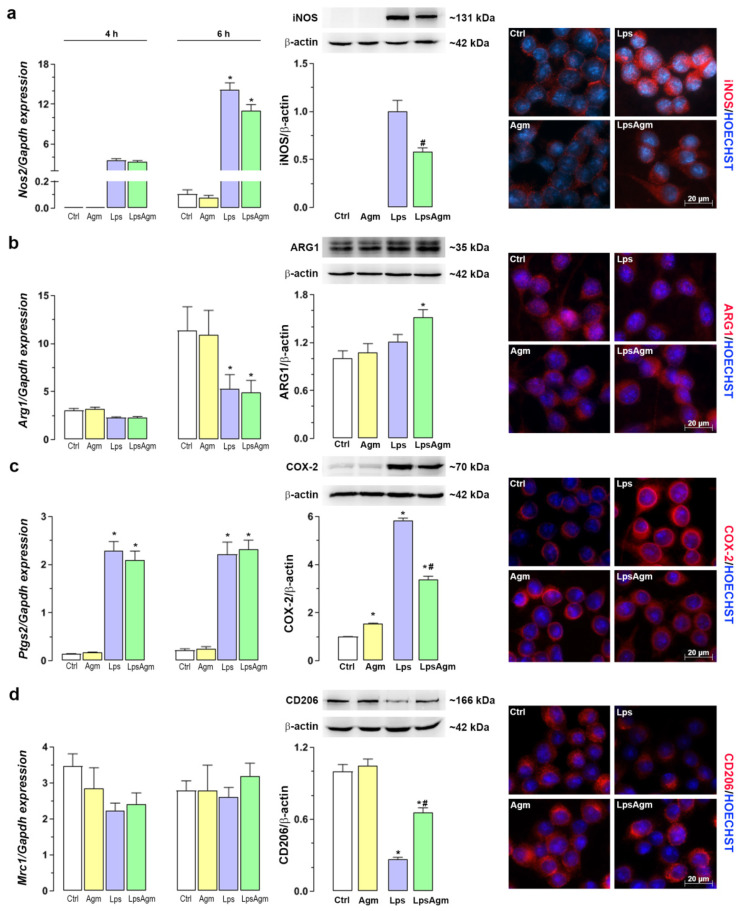
Effect of Agm on two main enzymes of arginine metabolism: iNOS and ARG1 and markers of a microglial activation state. BV-2 cells were pretreated with Agm (100 µm) for 30 min and then stimulated with Lps (1 µg/mL) for an additional 4 or 6 h (qPCR analysis) or 24 h (Western blot analysis and fluorescent microscopy). Data represent gene expressions, protein levels, and immunofluorescence of iNOS (**a**), ARG1 (**b**), COX-2 (**c**), and CD206 (**d**). Levels of target genes (*Nos2*, *Arg1*, *Ptgs2*, and *Mrc1*) are expressed relative to the expression of the *Gapdh* gene. The ratios of iNOS, ARG1, COX2, and CD206 to β-actin are expressed relative to the control group (fold change). Data are presented as mean ± SEM, from three separate determinations. The results were analyzed by Kruskal–Wallis test followed by Dunn’s multiple comparisons tests (qPCR) or by two-way ANOVA followed by Bonferroni’s post hoc tests (Western blot). iNOS protein level was analyzed by t test for comparing the mean values between two groups (Lps vs. LpsAgm) because Ctrl and Agm groups were not detectable. * *p* < 0.05 compared with the control group; # compared with Lps group. Photomicrographs on the right represent immunofluorescence labeling against target protein (red) counterstained with Hoechst (blue). If not otherwise specified, in this and following figures experimental groups are: Ctrl—untreated control cells; Agm—agmatine sulfate (100 µm) treated cells; Lps—cells stimulated with 1 µg/mL Lps; LpsAgm—cells pretreated with 100 µm Agm and then stimulated with Lps. Plotted scale bar applies to all micrographs.

**Figure 3 ijms-23-03561-f003:**
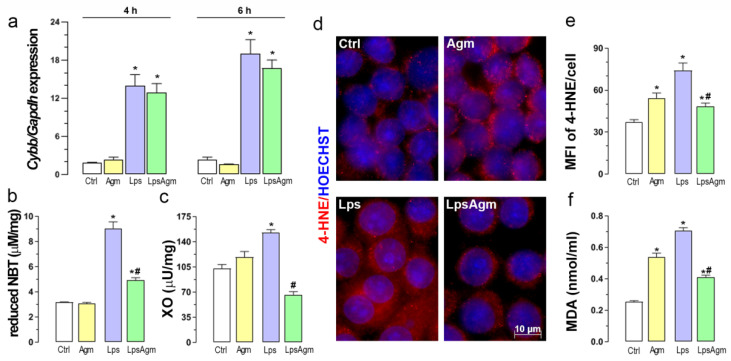
Effect of 100 µm Agm on O_2_^−^ level, XO activity, and lipid peroxidation in control and BV-2 cells stimulated with Lps. BV-2 cells were pretreated with Agm (100 µm) for 30 min and then stimulated with Lps (1 µg/mL) for an additional 4 and 6 h (qPCR analysis) or 24 h (O_2_^−^, XO, 4-HNE, and MDA). (**a**) NOX2 gene (*Cybb*) expression relative to the expression of the *Gapdh* gene. Superoxide-producing (**b**) and XO activity (**c**). Panel (**d**) represents immunofluorescence labeling against 4-HNE bound proteins (red) counterstained with Hoechst (blue); plotted scale bar applies to all micrographs. (**e**) Mean per-pixel fluorescence intensity (MFI) of 4-HNE per cell was measured in ~200 cells per experimental group and analyzed as described in Materials and Methods. (**f**) The intracellular concentration of MDA. Data collected from all assays are presented as mean ± SEM, from three separate determinations, and were analyzed by Kruskal–Wallis test followed by Dunn’s multiple comparisons tests. * *p* < 0.05 compared with a control group; # compared with Lps group.

**Figure 4 ijms-23-03561-f004:**
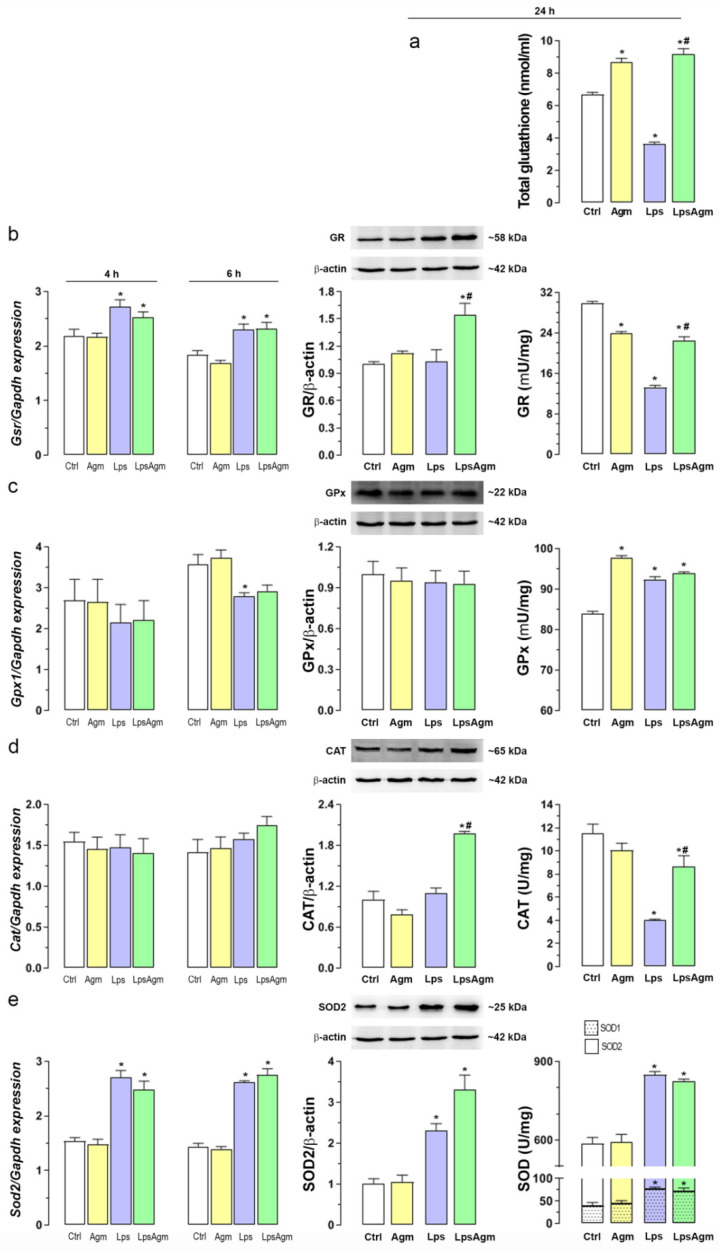
Effect of Agm on antioxidant stress response in Lps-stimulated BV-2 cells. Schedule of Lps stimulation: 4 and 6 h for qPCR analysis; 24 h for glutathione content, Western blot, and enzyme activity analyses. (**a**) Intracellular content of total glutathione. (**b**–**e**) Gene expression, protein levels, and enzyme activity of GR (**b**), GPx (**c**), CAT (**d**), and SODs (**e**). Levels of target genes are expressed relative to the expression of the *Gapdh* gene. The ratio of antioxidative protein/β-actin is expressed relative to the control group (fold change). Data are presented as mean ± SEM, from three to four separate determinations. Results were analyzed by: (i) Kruskal–Wallis test followed by Dunn’s multiple comparisons tests (*Gpx1*, *Cat*, *Sod2* gene, and GR, CAT, and SODs activity analyses); (ii) or by two-way ANOVA followed by Bonferroni’s post hoc tests (total glutathione level, *Gsr* gene, Western blot, and GPx activity analyses). Significance level: * *p* < 0.05 compared with control group; # compared with Lps group.

**Figure 5 ijms-23-03561-f005:**
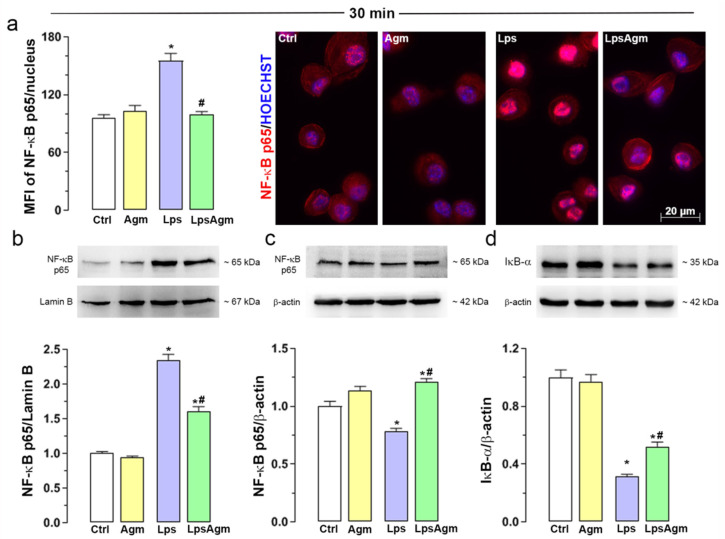
Effect of Agm on Lps-induced nuclear translocation of NF-κB p65. For nuclear and cytosolic extracts BV-2 cells were seeded at a density of 4.0 × 10^4^ cells/cm^2^. (**a**) The mean per-pixel fluorescence intensity (MFI) of the NF-κB p65 subunit in the nucleus of BV-2 cells was measured in ~200 cells per experimental group. Immunofluorescence labeling against p65 (red) and Hoechst fluorescence labeling (blue); plotted scale bar applies to all micrographs. (**b**) Nuclear protein levels of NF-κB p65 relative to lamin B protein content 30 min upon Lps stimulation. (**c**) NF-κB p65 levels in the cytosolic compartment. (**d**) Levels of IκB-α. Ratios of p65/lamin B or p65/β-actin, IκB-α/β-actin (nuclear or cytosolic expression, respectively) are expressed relative to the control (fold change) ± SEM from n = 4 separate determinations (a representative blot is shown). Data were analyzed by Kruskal–Wallis test followed by Dunn’s multiple comparisons tests (MFI of the NF-κB p65/nucleus) or two-way ANOVA followed by Bonferroni’s post hoc tests (Western blot analysis). Significance level: * *p* < 0.05 compared with the control group; # compared with Lps group.

**Figure 6 ijms-23-03561-f006:**
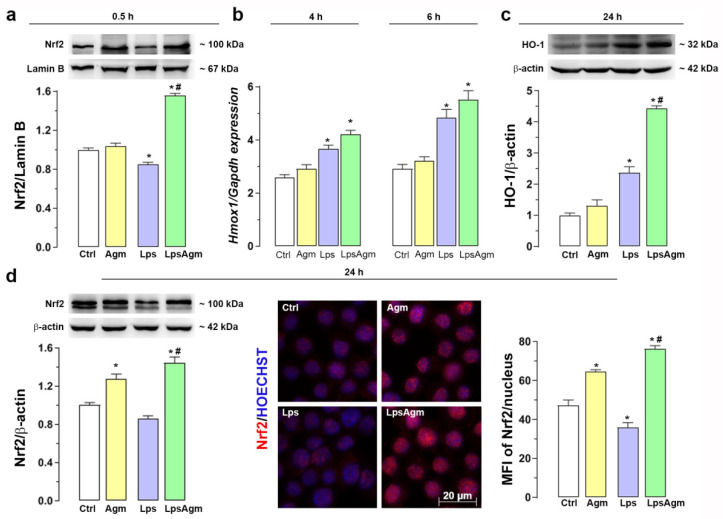
Agm effect on the transcription factor Nrf2 and antioxidant enzyme HO-1 expressions. For nuclear extracts BV-2 cells were seeded at a density of 4.0 × 10^4^ cells/cm^2^. Cells were pretreated with Agm (100 µm) for 30 min and then stimulated with Lps (1 µg/mL) for additional 0.5 and 24 h (Western blot analysis and fluorescent microscopy) or 4 and 6 h (qPCR analysis). (**a**) Nuclear protein level of Nrf2 relative to lamin B protein content 0.5 h upon Lps stimulation. (**b**) HO-1 gene (*Hmox1*) expression relative to the expression of the *Gapdh* gene and, (**c**) protein level relative to β-actin. (**d**) Nrf2 Western blotting (relative to β-actin) 24 h after Lps administration in cell lysates—left; micrographs with immunofluorescence labeling against Nrf2 (red) and Hoechst fluorescence labeling (blue), plotted scale bar applies to all images—middle; mean per-pixel fluorescence intensity (MFI) of the Nrf2 subunit in the nucleus of BV-2 cells was measured in ~200 cells per experimental group and analyzed as described in Materials and Methods—right. Protein levels are expressed relative to the control group (fold change) ± SEM (n = 4, a representative blot is shown). Results were analyzed by two-way ANOVA followed by Bonferroni’s post hoc tests (Western blot and immunofluorescence analyses) or Kruskal–Wallis test followed by Dunn’s multiple comparisons tests (gene expression analysis). Significance level: * *p* < 0.05 compared with control group; # compared with Lps group.

**Figure 7 ijms-23-03561-f007:**
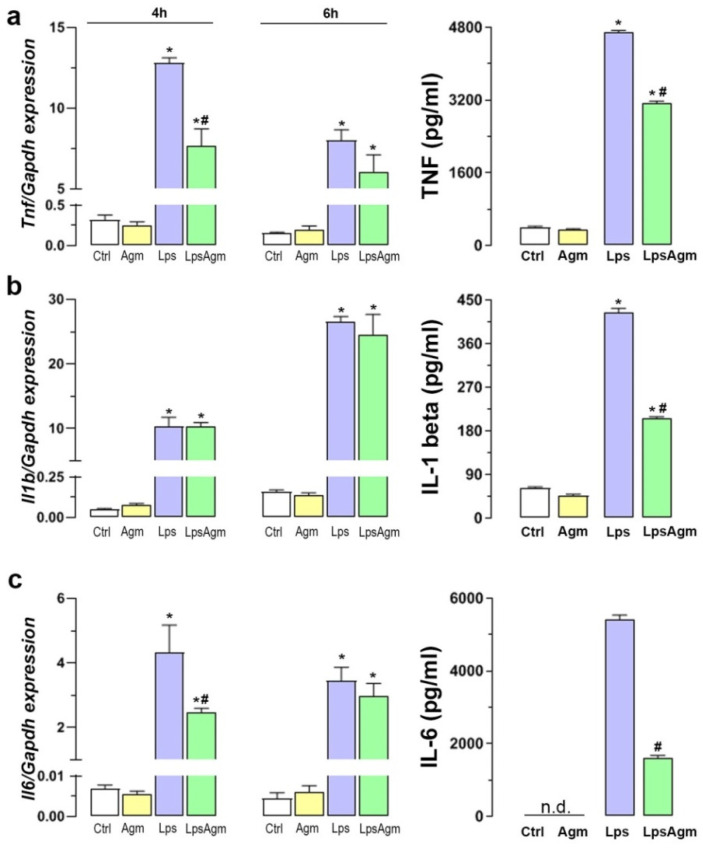
Effect of Agm on proinflammatory cytokines gene expression and release by BV-2 cells. Schedule of Lps stimulation: 4 and 6 h for qPCR analysis; 24 h for TNF, IL-1 beta, and IL-6 protein levels measured in cell supernatants by ELISA. (**a**–**c**) Gene expression and protein levels of TNF (**a**), IL-1 beta (**b**), and IL-6 (**c**) Data are presented as mean ± SEM from three separate determinations. Results were analyzed by two-way ANOVA followed by Bonferroni’s post hoc tests, except IL-6 secretion, which was analyzed by Mann–Whitney test for comparing the mean values between two groups (Lps vs. LpsAgm) because Ctrl and Agm groups were not detectable (labeled n.d. on (**c**)). Significance levels shown inside the graphs: * *p* < 0.05 compared with the control group; # compared with Lps group.

**Table 1 ijms-23-03561-t001:** Primer sequences used for qPCR analysis.

Primer	Sequence	Accession No.
*Gapdh*	f: GTTGTCTCCTGCGACTTCA r: TGGTCCAGGGTTTCTTACTC	NM_008084
*Nos2*	f: TTCACTCCACGGAGTAGCCT r: TGAGAACAGCACAAGGGGTT	NM_010927.4
*Arg1*	f: TAACCTTGGCTTGCTTCGG r: GTGGCGCATTCACAGTCAC	NM_007482.3
*Ptgs2*	f: TTCAACACACTCTATCACTGGC r: AGAAGCGTTTGCGGTACTCAT	NM_011198.4
*Mrc1*	f: GTTGTATTCTTTGCCTTTCCCAG r: CGTCTGAACTGAGATGGCACT	NM_008625.2
*Cybb*	f: GGGAACTGGGCTGTGAATGA r:CAGTGCTGACCCAAGGAGTT	NM_000397.4
*Gsr*	f: ACCGAGGAACTGGAGAATGC r: CACGGAAGTCACCACTTGGA	NM_010344.4
*Gpx1*	f: AGTCCACCGTGTATGCCTTCT r: GAGACGCGACATTCTCAATGA	NM_008160.6
*Cat*	f: AGCGACCAGATGAAGCAGTG r: TCCGCTCTCTGTCAAAGTGTG	NM_012520.2
*Sod2*	f: CAGACCTGCCTTACGACTATGG r: CTCGGTGGCGTTGAGATTGTT	NM_013671.3
*Hmox1*	f: CCTCACTGGCAGGAAATCATC r: CTCGTGGAGACGCTTTACATA	NM_010442.2
*Tnf*	f: GCCCACGTCGTAGCAAACCAC r: GGCTGGCACCACTAGTTGGTTGT	NM_013693.3
*Il1b*	f: AAAAGCCTCGTGCTGTCGGACC r: TTGAGGCCCAAGGCCACAGGT	NM_008361.4
*Il6*	f: TAGTCCTTCCTACCCCAATTTCC r: TTGGTCCTTAGCCACTCCTTC	NM_012589.2

**Table 2 ijms-23-03561-t002:** Antibodies used for Western blot (WB) and immunofluorescence (IF).

Antibody	Source and Type	Dilution	Manufacturer
iNOS	Rabbit, polyclonal	1:450 (IF) 1:500 (WB)	Abcam, ab15323
ARG1	Rabbit, polyclonal	1:100 (IF) 1:1000 (WB)	Sigma Aldrich, AV45673
COX-2	Goat, polyclonal	1:200 (IF) 1:1000 (WB)	Santa Cruz, sc-1745
CD206	Rabbit, polyclonal	1:500 (IF) 1:1000 (WB)	Abcam, ab64693
4-HNE	Rabbit, polyclonal	1:100 (IF)	Abcam, ab46545
GR	Rabbit, polyclonal	1:2500 (WB)	Abcam, ab16801
GPx	Rabbit, polyclonal	1:5000 (WB)	Abcam, ab22604
CAT	Rabbit, polyclonal	1:8000 (WB)	Sigma Aldrich, 219010
SOD2	Rabbit, polyclonal	1:5000 (WB)	Abcam, ab13533
NF-kB p65	Rabbit, polyclonal	1:200 (IF) 1:2000 (WB)	Santa Cruz, sc-372
IkB-α	Rabbit, polyclonal	1:1000 (WB)	Santa Cruz, sc-371
Nrf2	Rabbit, polyclonal	1:200 (IF) 1:1000 (WB)	Santa Cruz, sc-722
HO-1	Goat, polyclonal	1:1000 (WB)	Santa Cruz, sc-1796
β-actin	Mouse, monoclonal	1:5000 (WB)	Sigma Aldrich, A5316
Lamin B	Goat, polyclonal	1:1000 (WB)	Santa Cruz, sc-6217
Anti-rabbit IgG AlexaFluor 568	Donkey	1:250 (IF)	Invitrogen, A10042
Anti-rabbit IgG AlexaFluor 555	Donkey	1:250 (IF)	Invitrogen, A31572
Anti-rabbit IgG-HRP	Goat	1:5000 (WB)	Santa Cruz, sc-2004
Anti-rabbit IgG-HRP	Donkey	1:5000 (WB)	Santa Cruz, sc-2313
Anti-mouse IgG-HRP	Donkey	1:5000 (WB)	Santa Cruz, sc-2314
Anti-goat IgG-HRP	Donkey	1:5000 (WB)	Santa Cruz, sc-2020

## Data Availability

Not applicable.

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
