# Peer review of "Agmatine Mitigates Inflammation-Related Oxidative Stress in BV-2 Cells by Inducing a Pre-Adaptive Response"

_ijms, 2022, doi:10.3390/ijms23073561_

Round 1

Reviewer 1 Report

Broad comments

The manuscript is well organized and clearly easy to be read.
The research question is simple as well as nicely investigated and supported by different common biochemical techniques.
A limitation can be identified in the non-human in vitro microglia model that has been employed, however, the presented results are believed as very interesting and authors’ hypothesis has been rigorously tested.
The manuscript partially deserves its publication, in line within the aim and scopes of IJMS, but a mild revision should be necessary, both to ameliorate the quality to high standard of the journal chosen for submission.

Specific requests and comments

  1. Figure 6D double bands of NRF2 should be explained. Even the pattern is similar to the LPS stimulated condition and is justified by LPS exposition, why only the upper band is augmented in comparison to the lower one? may you say that the upper band was associated to Nrf2 nuclear localization (as confirmed in the side immunofluorescence panel and fig 6 panel a?
  2. Figure 7D, if the levels of IL-6 were undetectable, this should be stated/explained in both results and Fig. 7 legend, otherwise the used graphical representation can be ambiguous or seeming as lacking. This is important also because * p < 0.05 was accordingly not assigned in absence of Ctrl number.
  3. Since protein level of IL-6 were currently unknown (at least as this manuscript version shows), a qPCR for IL-6 might be performed. Why has not this been done? If major revision is requested by at the end of first round of revisions by the editors, please try to address this comment experimentally. In case a minor revision, please add a reply and some comment in the discussion of the paper.  
  4. Why in the abstract examples such as “iNOS and cyclooxygenase-2 levels” (instead of COX2) and “CD206 and heme oxygenase-1 levels” (instead of HO1) mixed abbreviation and not abbreviations for proteins/enzymes? Please make all similar
  5. Line 144, minor typo, “andMrc1” , please divide words
  6. A separate section titled “Abbreviation list” or fist-appearance extended name should be added for acronyms and molecules.
    The reason it that, for example MDA or 4-HNE (and many others), abbreviations should be extended soon as they are read, but now they are extended in section 4.5 at the very last part (Materials and Methods part in IJMS) of the manuscript, being currently less useful for not experts in chemistry and general audience
  7. Permeabilization” mentioned in section 4.5 (line 490) …whit what detergent (Triton/ Tween/etc.?) or other chemical? excuse the curiosity. Please add this detail to increase experimental reproducibility.
  8. Besides some endogenous mammalian regulators such as agmatine (Agm) the authors should mention in the discussion or introduction that also some phytochemicals, such as from agri-food waste, can present a certain ability to restrain inflammatory markers in LPS stimulated BV-2 model of overactivated microglia, with concentrations of endotoxin similar to the one used by the authors (please, see the following article Chiocchio et al., 2020, located at doi.org/10.3390/metabo10100408 , for useful and suggested reference)  .

Reviewer 2 Report

The work of Milosevic et al. deals with the possible mitigation of oxidative stress in LPS-induced BV2 cells by agmatine. The manuscript is well-written, showing interesting data that confirm previous experiments and new results. Some questions should be addressed before publication.

- Why is not two-way ANOVA used in all experiments? If the times are analyzed separately, they are experiments with two factors. On the other hand, do 4 and 6 h compare to each other? It would be interesting to know if there are differences. For example, in Fig. 2b, there seems to be a different expression of Arg1 at 4 and 6 h even in the control and Agm groups. Is this significant? If so, why? What does it mean?

- N= 3 does not seem very high, in addition to the technical and conceptual problems associated with the definition of N in cell culture. What is the power of statistical analysis throughout the manuscript? It should be included.

- IL-1beta is a key cytokine in the inflammatory response. Functionally speaking, its expression is activated by TNF and, in turn, activates that of IL-6. Therefore, the measure of this cytokine must be incorporated into this work.

- Although still widely used, there are some terms that are obsolete and should be replaced (as TNF-alpha, which should be just TNF) or discussed in a more current way, such as the so-called classic glial activation phenotypes (M1, lines 66, 396, 398) and alternative (M2, lines 80, 396,398). In relation to the term quiescent (defined by the Oxford Dictionary as quite, not active, dormant), although it is still used in the literature, I would like to indicate that in the opinion of many authors, it does not show the reality of microglial activity, cells that exert a permanent surveillance of the brain parenchyma. It may be more appropriate to use the term homeostatic. If one wishes to use the most current terminology, the idea that microglia switch between inactive and active states should be avoided.

- The use of hours or h should be homogenized.

- Line 120. The authors probably mean gene expression.

Round 2

Reviewer 2 Report

The authors have addressed all the questions raised. Regarding the calculation of power, the authors should be careful not to reach definitive conclusions based on those parameters that do not reach a power of 0.8; fortunately, there are only four parameters with inadequate power, three of which are supported by other parameters. The viability measured by MTT is the only one that is a bit unsupported, although I must say that this assay has never been one of my favorites for measuring cell viability. I think the authors could include this table as supplementary material. It has been a pleasure to participate in the revision of this work. For my part, I consider the manuscript to be ready for publication. Best wishes.